# Non-Invasive Assessment of Right Ventricle to Arterial Coupling for Prognosis Stratification of Fibrotic Interstitial Lung Diseases

**DOI:** 10.3390/jcm11206115

**Published:** 2022-10-17

**Authors:** Ciro Santoro, Agostino Buonauro, Angelo Canora, Gaetano Rea, Mario Enrico Canonico, Roberta Esposito, Alessandro Sanduzzi, Giovanni Esposito, Marialuisa Bocchino

**Affiliations:** 1Department of Advanced Biomedical Sciences, Federico II University, 80131 Naples, Italy; 2Respiratory Medicine Unit at the Monaldi Hospital, AO dei Colli, Department of Clinical Medicine and Surgery, Federico II University, 80131 Naples, Italy; 3Department of Radiology, Monaldi Hospital, AO dei Colli, 80131 Naples, Italy; 4Department of Clinical Medicine and Surgery, Federico II University Hospital, 80131 Naples, Italy

**Keywords:** right ventricle strain, right ventricular arterial coupling, echocardiography, pulmonary hypertension, fibrotic interstitial lung diseases, idiopathic pulmonary fibrosis

## Abstract

Background: The coupling of the right ventricle (RV) to the pulmonary circulation is an indicator of RV performance that can be non-invasively estimated by echocardiography. There are no data about its use in patients affected by fibrotic interstitial lung diseases (f-ILD). Methods: Fifty f-ILD patients, including 27 cases with idiopathic pulmonary fibrosis (IPF) (M = 37; mean age 67 ± 7 years), were studied with standard and speckle-tracking echocardiography and compared with 30 age-matched healthy volunteers. The mean patient follow-up was 70 ± 4 months. Results: Fibrotic ILD patients had a larger right ventricle (RV) and worse diastolic function because the RV global longitudinal strain (GLS) was significantly lower and the systolic pulmonary artery pressure (sPAP) estimates were higher in comparison with those of controls. Conversely, tricuspid annular systolic excursion (TAPSE) did not differ between controls and patients. Median values of TAPSE/sPAP and RV GLS/sPAP were significantly reduced in f-ILD patients (*p* < 0.0001). Patients with an RV GLS/sPAP below the median value had a shorter survival time (61 vs. 74 months, *p* = 0.01); this parameter was an independent predictor of a worse outcome. Conclusion: Low estimates of RV GLS/sPAP are predictive of worse outcomes in f-ILD patients. RV coupling seems to be a promising surrogate biomarker of RV performance to discriminate the patient phenotype with significant management and prognosis implications.

## 1. Introduction

Interstitial lung diseases (ILDs) include a heterogenous group of mainly chronic disorders, which are primarily characterized by alveolar and interstitial inflammation and/or fibrosis occurring either as an idiopathic entity or in response to different stimuli/associated with known conditions. Although they have different origins, ILDs share main pathophysiological mechanisms that give them common clinical presentations and lung function abnormalities. The most typical and disabling symptom is represented by reduced tolerance to physical effort with the progressive development of dyspnea and limitation of daily activities. Among the idiopathic ILDs, idiopathic pulmonary fibrosis (IPF) represents an entity with pure fibrotic imprinting burdened by an inexorable, rapid progression and a worse prognosis. Nevertheless, inflammation-driven ILDs, including connective tissue disease-associated ILDs (CTD-ILDs), chronic hypersensitivity pneumonitis (CHP), desquamative interstitial pneumonitis (DIP), and others, can acquire a fibrosing phenotype, which may lead to accelerated lung function decline and tissue distortion, similar to IPF [1,2].

Beyond causing parenchymal damage, ILDs can have a detrimental effect on the integrity and function of the pulmonary vasculature, thus aggravating exercise performance. Hypoxemia-induced vasoconstriction and pulmonary capillary stiffness contribute to the involvement of the right heart, whose increased afterload hesitates in the development of pulmonary hypertension (PH). This event has a significant impact on the disease’s natural history, as the ability of the right ventricle (RV) to adapt to the pressure increase in pulmonary circulation is a major determinant of ILD outcomes [3]. The coupling of the RV to the pulmonary circulation is representative of RV functional performance and correlates with the 18F-fuorodeoxyglucose (FDG) standardized uptake value (SUV), as measured by positron emission tomography (PET) imaging [4]. Therefore, RV uncoupling should be considered an early marker of RV failure and has potential applications for screening and monitoring purposes of lung diseases characterized by a chronic and progressive increase in systolic pulmonary arterial pressure (sPAP). The gold standard tool that allows the direct quantification of the RV-to-arterial coupling to be obtained is the invasive measurement of the pressure–volume loop-derived end-systolic ventricular and arterial elastance ratio (Ees/Ea). However, more recent studies have shown that this parameter can be indirectly estimated by non-invasive echocardiographic measurement as the ratio of RV-related parameters, which are either the tricuspid annular systolic excursion (TAPSE) or the RV global longitudinal strain (GLS), with the sPAP [5,6]. This last approach has significantly facilitated the use of RV–arterial coupling as a surrogate biomarker of RV performance in daily clinical practice, especially in light of its proven independent association with a high risk of mortality [7]. We retrospectively analyzed RV-to-arterial coupling, measured either as TAPSE/sPAP or RV GLS/sPAP, in patients affected by fibrotic ILDs (f-ILDs) enrolled at the time of first observation and undergoing echocardiographic evaluation as part of their diagnostic work-up at a tertiary referral center. The correlations of RV–arterial coupling with lung function parameters and the estimation of its prognostic value were assessed as well.

## 2. Materials and Methods

### 2.1. Study Population

The study population included 50 consecutive patients affected by f-ILDs who were referred to the respiratory diseases unit of Federico II University at Monaldi Hospital of Naples (Italy) between January 2015 and December 2016. Twenty-seven patients were affected by IPF and 23 were affected by f-ILDs other than IPF, including 10 patients affected by fibrotic nonspecific interstitial pneumonia (f-NSIP), 5 patients affected by CHP, 5 patients affected by DIP, and 3 patients affected by stage IV pulmonary sarcoidosis, were enrolled at the time of first observation and compared with 30 age- and sex-matched healthy subjects. With reference to IPF patients, diagnoses were made according to the 2011 international criteria [8]. No patients were undergoing anti-fibrotic therapy at enrolment. The anti-fibrotic drugs nintedanib and pirfenidone were prescribed only to eligible IPF patients (in 11 and 16 cases, respectively) for their disease-specific indication. Patients with f-ILDs underwent conventional treatments with systemic corticosteroids (prednisone) or, in selected cases, immune-suppressive drugs. The exclusion criteria were coronary artery disease and previous myocardial infarction, more than mild valvular heart disease, congestive heart failure, primary cardiomyopathies, atrial fibrillation, and congenital heart disease. Patients for whom the quality of echocardiography imaging was inadequate were also ruled out. Lung function assessment included spirometry, lung volume measurement, and determination of the hemoglobin (Hb)-adjusted single-breath diffusing lung capacity of carbon monoxide (DLCO_sb_). All tests were performed using a computer-assisted spirometer (Quark PFT 2008 Suite Version Cosmed Ltd., Rome, Italy) according to international standards [9,10,11]. The 6-min walk test (6-MWT) was performed in ambient air by trained hospital staff according to the guidelines [12]. The study was conducted in accordance with the amended Declaration of Helsinki and approved by the institutional ethical committee (protocol 1129/2015). All patients gave their written informed consent to participate in the study. Patients’ data were collected in a dedicated database in an anonymous way.

### 2.2. Echocardiographic Examination

All patients underwent a standard echo-Doppler exam at enrollment, including speckle tracking echocardiography (STE) for longitudinal strain analysis of both ventricles with a Vivid E95 ultrasound machine (GE Healthcare, Horten, Norway) using a 2.5 MHz transducer with harmonic capability according to the standards of our laboratory and the European Association of Cardiovascular Imaging [13,14,15,16]. Right ventricle diameters (transverse basal, mid-cavity, and longitudinal diameters) were measured at end-diastole in an apical 4-chamber view oriented to obtain the maximal RV internal chamber size. RV global systolic function was assessed by measuring both TAPSE (mm) by M-mode echo and systolic (s’) by tissue Doppler imaging (TDI) [17]; sPAP was extrapolated according to the guidelines on the basis of the peak of tricuspid regurgitation velocity (TRV) and the addition of an estimate of right atrial pressure (RAP) in apical 4-chamber view [18,19,20]. The RV GLS was determined by averaging values of the 6 segments: 3 from the free lateral wall and 3 from the interventricular septal wall [13,21]. The RV GLS values were considered positive (sign +) to strengthen the clinical meaning; the higher the value, the better the strain deformation (Figure 1).

RV-to-arterial coupling was estimated as the ratio between TAPSE or RV GLS and sPAP. Cut-off values ≤0.31 mm/mmHg or ≤0.35%/mmHg, respectively, were used to define RV uncoupling, as previously described [5,22]. Left chamber (size and function) quantification was assessed as previously described [17,19]. Left ventricle (LV)-STE acquisition was obtained in the three apical views (long-axis, 4-chamber, and 2-chamber) and RV-STE acquisition was obtained in the four apical views according to our laboratory standards [23,24]. Post-processing was performed offline on a dedicated workstation (EchoPAC software only version 204, GE Healthcare, Chicago, IL, USA). LV longitudinal strain was measured as the segmental peak strain before the aortic valve closure of 6 segments (basal, medium, and apical segments for each wall) in each of the three apical views (4-chamber, 2-chamber, and 3-chamber), and GLS was derived as the average of the peak systolic strain of the 17 segments, as recommended [23].

### 2.3. Statistical Analysis

Data are presented as absolute numbers (percentages), mean values ± standard deviation (SD), or medians and interquartile ranges (IQRs) [IQR25–IQR75], where appropriate. Statistical analysis was performed using the Student’s *t*-test for continuous data with a normal distribution or the Mann–Whitney test for non-parametric comparisons between groups. Spearman’s correlation test was used to evaluate univariate correlates of a given variable. Univariate and multivariate Cox proportional hazards regression analyses were performed to identify independent variables associated with the endpoint. In the multivariate Cox analysis, RV–arterial coupling values were multiplied by ten to enhance the clinical significance of 0.1 variation. Optimal cut-off values for RV coupling associated with the outcome were calculated by time-dependent receiver–operator characteristic (ROC) curves by calculating the area under the curves. Kaplan–Meier analysis was used to compare the differences in survival and was implemented with the log-rank test. The cumulative survival time was calculated as the time from the date of the first observation (first diagnosis) to the date of death or censorship on the date of the last follow-up visit. The null hypothesis was rejected at *p* ≤ 0.05. Statistical analysis was performed with SPSS version 21 (SPSS Inc., Chicago, IL, USA).

## 3. Results

### 3.1. Patient Characteristics

The demographics, clinical features, and main lung function parameters of the study population are reported in Table 1. Since the comparative analysis of the two subgroups of patients (27 patients with IPF vs. 23 patients with f-ILDs other than IPF) showed no significant differences with the exception of older age in IPF patients (69 vs. 66 years, *p* < 0.05), all of them were considered a unique group named f-ILDs. Thirty healthy subjects matched by age and sex, referred to our attention for voluntary cardiovascular screening, were recruited as controls. Compared with controls, f-ILD patients had a higher body mass index (BMI) (*p* < 0.001). Systemic arterial hypertension was the most prevalent comorbidity (60%) in f-ILD patients. Overall, lung function testing showed that f-ILD patients had a mild restrictive ventilatory pattern with a moderate deficiency of DLCO_sb_. None of the patients required continuous oxygen therapy at the time of study inclusion.

### 3.2. RV-to-Arterial Coupling in f-ILDs 

The echocardiography data of the most relevant RV-related parameters are reported in Table 2. Patients affected by f-ILDs had a larger ventricle than the controls, as shown by the finding of significantly higher values of basal (*p* < 0.0001), middle (*p* < 0.001), and longitudinal diameters (*p* = 0.02). In addition, patients displayed significantly lower RV GLS values (*p* < 0.0001) along with higher sPAP estimates (*p* < 0.0001) than controls. The median values of RV-to-arterial coupling computed by RV GLS to sPAP were significantly reduced in f-ILD patients compared with controls (0.55 vs. 0.92, *p* < 0.0001). Significant differences were also found in the median values of RV coupling when estimated as the TAPSE/sPAP ratio (0.60 vs. 0.84; *p* < 0.0001), although TAPSE values did not differ between patients and controls. RV-to-arterial coupling showed a significant direct correlation with patients’ DLCO_sb_ (r = 0.54, *p* < 0.0001) only when estimated as RV GLS/sPAP. Overall, there were no significant differences in the analysis of the two patient subgroups (IPF vs. ILDs other than IPF).

With reference to the LV echocardiography study parameters, no significant differences were recorded between f-ILD patients and controls with the exception of lower values of the E/A ratio in the former (*p* = 0.002) (Table 2).

### 3.3. Survival Analysis in f-ILDs

The study patients were followed for an average period of 70 ± 4 months. Patients affected by IPF had a worse prognosis than those affected by other ILDs (mean survival of 62 ± 5 vs. 73 ± 3 months, *p* = 0.02) (Figure 2A). To assess the impact of RV-to-arterial coupling on long-term survival, the entire sample of patients was divided according to its median value. As shown in Figure 2B, f-ILD patients with an RV GLS/sPAP below the median value had a significantly lower survival when compared with those with values above (61 ± 4 vs. 74 ± 3 months, *p* = 0.01). Conversely, when RV–arterial coupling was measured as TAPSE/sPAP, no significant differences (*p* = 0.17) in terms of survival were observed. 

In the univariate logistic regression analysis, arterial O_2_ partial pressure and DLCO_sb_ had significant associations with the endpoint (*p* < 0.009 and *p* < 0.02, respectively). The same was true for sPAP (*p* = 0.02) and RV–arterial coupling estimated as RV GLS/sPAP (*p* = 0.001), as reported in detail in Table 3. No significant association was found with LV-related echocardiographic parameters (data not shown). Multivariate stepwise regression analysis showed that IPF diagnosis and RV GLS/sPAP were the sole independent predictors of a worse outcome (Table 3).

## 4. Discussion

To the best of our knowledge, this is the first report that non-invasively assesses the RV-to-pulmonary circulation coupling by means of echocardiography in a cohort of f-ILD patients, including patients with idiopathic pulmonary fibrosis. Our data showed that the median values of RV-to-arterial coupling were significantly reduced, with no distinction between IPF patients and those affected by other ILDs. This observation was confirmed by estimating the RV–arterial coupling as both TAPSE/sPAP and RV GLS/sPAP. Interestingly, low values of RV–arterial coupling estimated as RV GLS/sPAP were associated with reduced survival and represented an independent predictor of a worse outcome.

There is a close inter-relationship between lung diseases and the right heart. This is particularly true with reference to f-ILDs, in which the distortion of the organ architecture is often associated with vascular bed damage and remodeling. In these cases, the concomitance of the resulting pulmonary hypertension and chronic RV overload are relevant predictive factors of increased mortality [25]. The prognosis stratification of patients affected by IPF or other f-ILDs mainly relies on the rate of decline of conventional lung function parameters. In light of the above considerations and our data, integrating the evaluation of these patients with the study of RV function can be of help for better phenotypic characterization. TAPSE and RV GLS are the main echocardiographic parameters used to assess RV systolic performance. TAPSE allows the estimation of the longitudinal excursion of the sole basal segment of the lateral wall toward the apex; it is assumed that such an excursion throughout the cardiac cycle is representative of the overall RV systolic function. However, this assumption may be fallacious, particularly in ventricles remodeled because of chronically increased afterload [26]. With reference to our study, the lack of significant differences in TAPSE values between patients and controls could account for a low accuracy of this parameter for prognosis stratification purposes, at least in relation to early RV dysfunction in the absence of signs of established PH. Certainly, speckle tracking analysis is a more accurate and advanced approach that better reflects the global RV contractility. There is evidence that the assessment of longitudinal strain by STE may help clinicians track changes in RV function with a relevant impact on disease management and outcome prediction [27,28]. In this regard, we previously reported that early reduction of RV strain was associated with clinical deterioration and was correlated with lung function impairment in patients affected by IPF [29]. As an emerging and attractive biomarker, RV-to-pulmonary circulation coupling is an indicator of the relationship between the RV contractility and its afterload; therefore, any perturbation of this interaction reflects early RV dysfunction, as previously shown in studies that invasive measured the pressure–volume loop-derived Ees/Ea ratio [30,31]. The possibility of measuring RV–arterial coupling with non-invasive methods is a privilege for the study of fragile patients, such as those included in our study. For this reason, we captured this opportunity by proposing the indexation of RV GLS with sPAP as a surrogate prognostication marker in f-ILDs on the basis of previous evidence [4,32]. Our choice seems to be promising because, although there was a significant reduction in TAPSE/sPAP in patients compared with controls, this more conventional estimate of the RV–arterial coupling did not allow a prognostic stratification. This observation confirms the lower performance of TAPSE in comparison with RV GLS in the early detection of right ventricular dysfunction, as previously discussed. In addition, although indirectly, it attributes greater accuracy to the measurement of arterial RV coupling in terms of RV GLS/sPAP, as also suggested by Houard et al. [33].

Iacovello M. et al., showed that RV–arterial uncoupling (for RV GLS/sPAP values < 0.35%/mmHg) was associated with increased mortality risk in 315 patients with heart failure and reduced ejection fraction (EF) [6]. Our findings partly agree with these data. The first difference may be that our population likely had mild to moderate pre-capillary instead of post-capillary PH, along with a preserved EF. Secondly, as only three of our study participants had an RV GLS/sPAP value below the reported cut-off of 0.35%/mmHg, patients’ survival curves were stratified according to the median value. Despite this methodological deviation, we observed a significant association with reduced survival in patients with values of RV GLS/sPAP below the median. ROC analysis also showed that the cut-off of 0.55%/mmHg was the best discriminator, with a sensitivity of 74% and a specificity of 63% (data not shown). The involvement of the pulmonary circulation in fibrotic ILDs and IPF has been neglected in some ways. This is likely due to the failure of clinical trials aimed at the treatment of PH in these patients. Although preliminary, our data could have applications regarding the selection of patients who can benefit most from such treatments. This hope is reinforced by the resumption of attempts in this direction [34].

Our study is limited by the single-center setting and small patient cohort, which do not allow definitive conclusions to be drawn. In addition, due to the retrospective explorative nature, no a priori formal hypothesis was formulated with a calculation of the sample size and study power. Overall, the impact of any alteration in RV–arterial coupling requires validation through further efforts in larger prospective populations with regard to both the different ILD categories (due to the multiple underlying mechanisms that can affect the RV performance in these patients) and the level of PH severity.

## 5. Conclusions

We showed that patients with f-ILDs had low RV–arterial coupling values, as estimated by the echocardiography-derived ratios of TAPSE/sPAP and RV GLS/sPAP. Interestingly, only the latter allowed prognosis stratification. The phenotypic characterization of f-ILD patients should be encouraged in multi-disciplinary settings. From this perspective, our findings open the way to an extremely interesting research field that may have a significant impact on disease management.

## Figures and Tables

**Figure 1 jcm-11-06115-f001:**
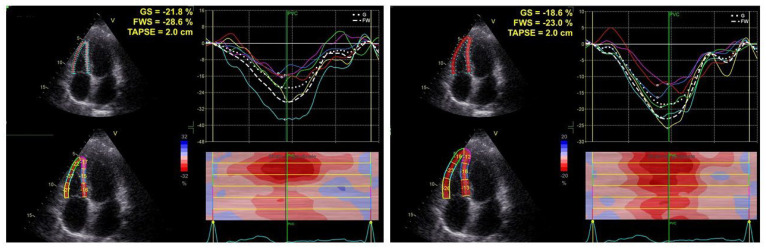
Representative images of right ventricle strain analysis in a healthy control (**left panel**) and in a patient affected by fibrotic interstitial lung disease (**right panel**).

**Figure 2 jcm-11-06115-f002:**
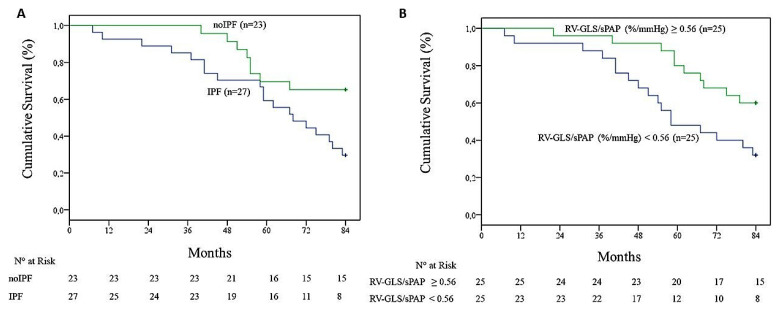
(**A**) Kaplan–Meier curves representative of the cumulative survival of idiopathic pulmonary fibrosis (IPF) patients vs. those with fibrotic interstitial lung diseases other than IPF (here reported as non-IPF). (**B**) Kaplan–Meier curves representative of the cumulative survival of the whole study population according to the right ventricle (RV) coupling to the pulmonary circulation, estimated as the ratio between the RV global longitudinal strain (RV GLS) and the systolic pulmonary artery pressure (sPAP).

**Table 1 jcm-11-06115-t001:** Demographics, clinical characteristics, and lung function of the study population.

Parameter	f-ILDs	Controls	*p*
	(*n* = 50)	(*n* = 30)	
*Demographics and clinical data*			
Age (years)	67 ± 7	67 ± 8	0.857
Sex (males)	37 (74)	21 (70)	
BMI (kg/m^2^)	28.9 ± 4.3	25.2 ± 3.1	**<0.001**
Smoking (never/former/smokers)	20 (40)/30 (60)/0 (0)	11 (37)/18 (60)/1 (3)	
Systemic arterial hypertension	30 (60)	5 (17)	
Chronic ischemic heart disease	7 (14)	0 (0)	
Type II diabetes	9 (18)	0 (0)	
Gastro-esophageal reflux	18 (36)	11 (37)	
*Lung function*			
paO_2_ (mmHg)	74 (67–83)	/	/
FVC (% pred)	61 (51–80)	/	/
TLC (% pred)	57 (46–73)	/	/
RV (% pred)	48 (39–91)	/	/
DLCO_sb_ (% pred)	47 (34–71)	/	/
6-MWT (meters)	407 (252–520)	/	/

Data are expressed as absolute numbers (percentage), mean values ± SD, or medians (IQR25–IQR75), where appropriate. Statistically significant results are reported in bold. Abbreviations: f-ILDs = fibrotic interstitial lung diseases; BMI = body mass index; SBP = systolic arterial pressure; DBP = systolic blood pressure; HR = heart rate; paO_2_ = arterial O_2_ partial pressure; FVC = forced vital capacity; TLC = total lung capacity; RV = residual volume; DLCO_sb_ = single-breath diffusing lung capacity for carbon monoxide; 6-MWT = 6-min walk test; SD = standard deviation; IQR = interquartile.

**Table 2 jcm-11-06115-t002:** Right and left heart echocardiographic evaluation in fibrotic interstitial lung diseases.

Parameter	f-ILDs	Controls	*p*
	(*n* = 50)	(*n* = 30)	
*Right heart*			
RV basal diameter (mm)	39 (36–43)	35 (31–36)	**0.0001**
RV middle diameter mm)	32 (28–36)	28 (24–31)	**0.03**
RV longitudinal diameter (mm)	63 (57–69)	59 (52–64)	0.1
RV TDI s’ (m/s)	0.13 (0.12–0.16)	0.13 (0.11–0.16)	0.85
TAPSE (mm)	22 (19–25)	23 (21–26)	0.27
TRV (m/s)	2.57 (2.49–2.75)	2.12 (2.04–2.30)	**0.001**
sPAP (mmHg)	35 (29–41)	26 (24–31)	**0.0001**
RV-GLS (%)	20 (19–22)	24 (22–27)	**0.0001**
TAPSE/sPAP (mm/mmHg)	0.60 (0.24–1.10)	0.84 (0.39–1.32)	**0.0001**
RV-GLS/sPAP (%/mmHg)	0.55 (0.47–0.67)	0.92 (0.77–1.07)	**0.0001**
*Left heart*			
IVSd (cm)	0.9 (0.8–0.95)	0.9 (0.85–1.0)	0.88
LVEDd (cm)	4.8 (4.5–5.0)	4.8 (4.5–5.1)	0.6
LVESd (cm)	2.9 (2.6–3.2)	3.0 (2.6–3.4)	0.65
RWT	0.37 (0.34–0.42)	0.36 (0.34–0.39)	0.78
LVMi (g/m^2^)	38 (33.3–46.1)	39 (31.5–44.1)	0.78
E/A ratio	0.74 (0.62–0.87)	0.85 (0.81–0.89)	**0.002**
DT (ms)	251 (209–273)	22 (195–292)	0.29
E/e’ ratio	8.5 (7.1–10.4)	7.3 (6.2–9.7)	0.29
LV-EF (%)	63 (60–67)	62.5 (58.5–64.7)	0.32
LV-GLS (%)	20.9 (19.0–22.7)	22.2 (20.3–23.7)	0.09

Data are expressed as medians (IQR25–IQR75). Statistically significant results are reported in bold. Abbreviations: f-ILDs = fibrotic interstitial lung diseases; RV = right ventricle; TDI = tissue Doppler imaging; TAPSE = tricuspid annular systolic excursion; TRV = tricuspid regurgitation velocity; sPAP = systolic pulmonary arterial pressure; GLS = global longitudinal strain; IVSd = interventricular septal diameter; LVEDd = left ventricular end-diastolic diameter; LVESd = left ventricular end-systolic diameter; RWT = relative wall thickness; LVMi = left ventricular mass index; DT = deceleration time; EF = ejection fraction.

**Table 3 jcm-11-06115-t003:** Logistic regression analysis.

**Variable**	**Univariate Analysis**	
	**Odds Ratio (95% CI)**	** *p* **
Age	1.01 (0.95–1.08)	0.69
Sex	2.50 (0.87–7.40)	0.09
BMI	0.97 (0.87–1.08)	1.59
*Lung function*		
paO_2_	0.93 (0.88–0.98)	**0.009**
FVC	0.99 (0.98–1.01)	0.84
RV	0.99 (0.98–1.01)	0.63
TLC	1.00 (0.99–1.01)	0.33
DLCO_sb_	0.97 (0.95–0.99)	**0.02**
6-MWT	1.00 (0.99–1.00)	0.31
*RV parameter*		
sPAP	1.05 (1.01–1.09)	**0.02**
RV-GLS	0.84 (0.65–1.08)	0.17
RV-GLS/sPAP	0.60 (0.44–0.81)	**0.001**
*Other*		
IPF diagnosis	2.51 (1.10–5.74)	**0.03**
**Variable**	**Multivariate Analysis**	
	**Odds Ratio (95% CI)**	** *p* **
IPF diagnosis	2.20 (0.95–5.09)	**0.04**
RV-GLS/sPAP	0.62(0.46–0.83)	**0.002**

Abbreviations: BMI = body mass index; paO_2_ = arterial O_2_ partial pressure; FVC = forced vital capacity; RV = residual volume; TLC = total lung capacity; DLCO_sb_ = single-breath diffusing lung capacity for carbon monoxide; 6-MWT = 6-min walk test; sPAP = systolic pulmonary arterial pressure; GLS = global longitudinal strain.

## Data Availability

Data available on request due to privacy reasons, available from the corresponding author.

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
