# Peer review of "Non-Invasive Assessment of Right Ventricle to Arterial Coupling for Prognosis Stratification of Fibrotic Interstitial Lung Diseases"

_jcm, 2022, doi:10.3390/jcm11206115_

Round 1

Reviewer 1 Report

I have two methodological remarks:

1. Why did the Authors choose for the RV-PA coupling evaluation global RV strain ? The free RV strain / sPAP parameter is now considered to be more appropriate. I suggest to eveluate both.  

2. In Table 2 the presence and the degree of the functional TR should be mentioned as it may influence TAPSE, which in turn may make TAPSE/sPAP  parameter less useful in prognostic assessment. 

In the abstract in line 20 should be"worst systolic function" (not diastolic).

Author Response

  1. To our knowledge, the validation of RV strain/sPAP parameter has not yet been archived. Additionally, there isn't yet a specific study comparing the global RV strain to the free wall strain. We cited papers where both parameters were tested. In our study we assumed that myocardial injury affected the entire RV (including the septal and lateral walls), so we essentially used global RV strain as a measure of impaired longitudinal function.
  2. We mentioned the frequency of functional tricuspid regurgitation due to primary RV enlargement, in the results section. However, patients affected by moderate to severe tricuspid regurgitation or coronary artery disease were excluded as specified in methods section.
  3. We thank the reviewer for the suggestion. We modified the text accordingly

Reviewer 2 Report

Overall I really liked the manuscript. RV-Arterial coupling estimation is very useful in many RV oriented diseases and in my center - especially in PAH. Thus, I suggest adding few references about MRI/echo/RHC estimation of Tapse/spap in PAH population as it is also RV oriented group. 
I also suggest adding ROC analysis for cut off value if possible. 

Author Response

  1. We thank the reviewer for the comment. We concur that RV strain/sPAP is an emerging parameter for RV-PA coupling assessment. Moreover, an updated reference on RV strain/sPAP assessed by MRI and PAH was mentioned in the text (Tello et al. International Journal of Cardiology 2018) Finally, we reported ROC analysis in the results section (pag 13 from lines 8 to 10).